# SOLOv2: Dynamic and Fast Instance Segmentation

**Xinlong Wang**[1]     **Rufeng Zhang**[2]     **Tao Kong**[3]     **Lei Li**[3]     **Chunhua Shen**[1]

[1]The University of Adelaide, Australia     [2]Tongji University, China     [3]ByteDance AI Lab

## Abstract

In this work, we design a simple, direct, and fast framework for instance segmentation with strong performance. To this end, we propose a novel and effective approach, termed SOLOv2, following the principle of the SOLO method [32]. First, our new framework is empowered by an efficient and holistic instance mask representation scheme, which dynamically segments each instance in the image, without resorting to bounding box detection. Specifically, the object mask generation is decoupled into a mask kernel prediction and mask feature learning, which are responsible for generating convolution kernels and the feature maps to be convolved with, respectively. Second, SOLOv2 significantly reduces inference overhead with our novel matrix non-maximum suppression (NMS) technique. Our Matrix NMS performs NMS with parallel matrix operations in one shot, and yields better results. We demonstrate that the proposed SOLOv2 achieves the state-of-the-art performance with high efficiency, making it suitable for both mobile and cloud applications. A light-weight version of SOLOv2 executes at 31.3 FPS and yields 37.1% AP on COCO `test-dev`. Moreover, our state-of-the-art results in object detection (from our mask byproduct) and panoptic segmentation show the potential of SOLOv2 to serve as a new strong baseline for many instance-level recognition tasks. Code is available at `https://git.io/AdelaiDet`

## 1 Introduction

Generic object detection aims at localizing individual objects and recognizing their categories. For representing the object locations, bounding box stands out for its simplicity. Localizing objects using bounding boxes have been extensively explored, including the problem formulation, network architecture, post-processing and all those focusing on optimizing and processing the bounding boxes. The tailored solutions largely boost the performance and efficiency, thus enabling wide downstream applications recently. However, bounding boxes are coarse and unnatural. Human vision can effortlessly localize objects by their irregular boundaries. Instance segmentation, *i.e.*, localizing objects using masks, pushes object localization to the limit at pixel level and opens up opportunities to more instance-level perception and applications. To date, most existing methods deal with instance segmentation in the view of bounding boxes, *i.e.*, segmenting objects in (anchor) bounding boxes. How to develop pure instance segmentation including the supporting facilities, *e.g.*, post-processing, is largely unexplored compared to bounding box detection and instance segmentation methods built on top it.

We are motivated by the recently proposed SOLO framework (Segmenting Objects by LOcations) [32]. SOLO formulates the task of instance segmentation as two sub-tasks of pixel-level classification, solvable using standard FCNs, thus dramatically simplifying the formulation of instance segmentation. It takes an image as input, directly outputs instance masks and corresponding class probabilities, in a fully convolutional, box-free and grouping-free paradigm. However, three main bottlenecks limit the performance of SOLO: a) inefficient mask representation and learning; b) not high enough resolution for finer mask predictions; c) slow mask NMS. In this work, we eliminate the above bottlenecks all at once.

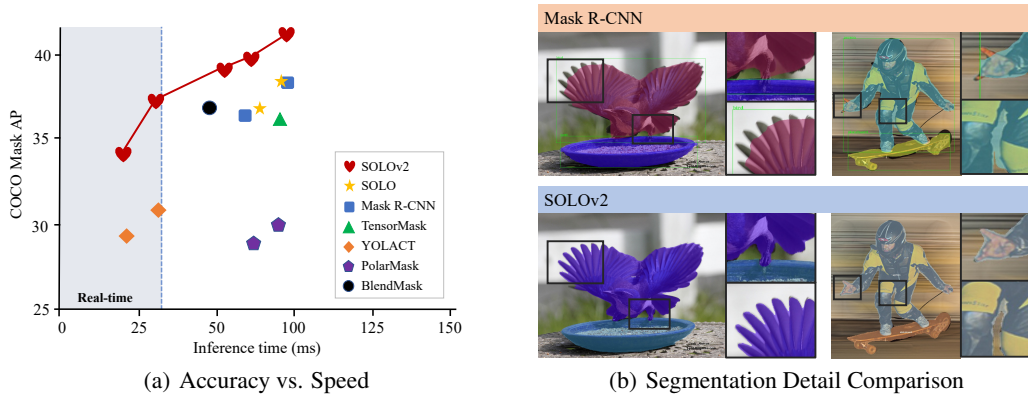

|     |     |
| --- | --- |
| (a) Accuracy vs. Speed | (b) Segmentation Detail Comparison |

**Figure 1** – Comparison of instance segmentation performance by SOLOv2 and other methods on the COCO `test-dev`. (a) The proposed SOLOv2 outperforms a range of state-of-the-art algorithms. All methods are evaluated using one Tesla V100 GPU. (b) SOLOv2 obtains higher-quality masks compared with Mask R-CNN. Mask R-CNN's mask head is typically restricted to $28 \times 28$ resolution, leading to inferior prediction at object boundaries.

We first introduce a dynamic scheme, which enables dynamically segmenting objects by locations. Specifically, the mask learning process can be divided into two parts: convolution kernel learning and feature learning (Figure 2(b)). When classifying the pixels into different location categories, the mask kernels are predicted dynamically by the network and conditioned on the input. We further construct a unified and high-resolution mask feature representation for instance-aware segmentation. As such, we are able to predict high-resolution object masks, as well as learning the mask kernels and mask features separately and efficiently.

We further propose an efficient and effective matrix NMS algorithm. As a post-processing step for suppressing the duplicate predictions, non-maximum suppression (NMS) serves as an integral part in state-of-the-art object detection systems. Take the widely adopted multi-class NMS for example. For each class, the predictions are sorted in descending order by confidence. Then for each prediction, it removes all other highly overlapped predictions. Such sequential and recursive operations result in non-negligible latency. For mask NMS, this drawback is further magnified. Compared to bounding box, it consumes more time to compute the IoU of each mask pair, thus leading to huge overhead. We address this problem by introducing Matrix NMS, which performs NMS with parallel matrix operations in one shot. Our Matrix NMS outperforms the existing NMS and its varieties in both accuracy and speed. As a result, *Matrix NMS processes 500 masks in less than 1 ms in simple python implementation*, and outperforms the recently proposed Fast NMS [2] by 0.4% AP.

With these improvements, SOLOv2 outperforms SOLO by 1.9% AP while being 33% faster. The Res-50-FPN SOLOv2 achieves 38.8% mask AP at 18 FPS on the challenging MS COCO dataset, evaluated on a single V100 GPU card. A light-weight version of SOLOv2 executes at 31.3 FPS and yields 37.1% mask AP. Interestingly, although the concept of bounding box is thoroughly eliminated in our method, our bounding box byproduct, *i.e.*, by directly converting the predicted mask to its bounding box, yields 44.9% AP for object detection, which even *surpasses many state-of-the-art, highly-engineered object detection methods*.

We believe that, with our simple, fast and sufficiently strong solution, instance segmentation can be a popular alternative to the widely used object bounding box detection, and SOLOv2 may play an important role and predict its wide applications.

## 1.1 Related Work

**Instance segmentation.** Instance segmentation is a challenging task, as it requires instance-level and pixel-level predictions simultaneously. The existing approaches can be summarized into three categories. Top-down methods [20, 12, 25, 14, 6, 2, 4, 38] solve the problem from the perspective of object detection, *i.e.*, detecting first and then segmenting the object in the box. In particular, recent methods of [4, 38, 35] build their methods on the anchor-free object detectors [31], showing promising performance. Bottom-up methods [27, 9, 24, 10] view the task as a label-then-cluster

problem, *e.g.*, learning the per-pixel embeddings and then clustering them into groups. The latest direct method (SOLO) [32] aims at dealing with instance segmentation directly, without dependence on box detection or embedding learning. In this work, we appreciate the basic concept of SOLO and further explore the direct instance segmentation solutions.

We specifically compare our method with the recent YOLACT [2]. YOLACT learns a group of coefficients which are normalized to $(-1, 1)$ for each anchor box. During the inference, it first performs a bounding box detection and then uses the predicted boxes to crop the assembled masks. While our method is evolved from SOLO [32] through directly decoupling the original mask prediction into kernel learning and feature learning. *No anchor box is needed. No normalization is needed. No bounding box detection is needed.* We directly map the input image to the desired object classes and object masks. Both the training and inference are much simpler. As a result, our proposed framework is much simpler, yet achieving significantly better performance (6% AP better at a comparable speed); and our best model achieves 41.7% AP vs. YOLACT's best 31.2% AP.

**Dynamic convolutions.** In traditional convolution layers, the learned convolution kernels stay fixed and are independent on the input, *i.e.*, the weights are the same for arbitrary image and any location of the image. Some previous works explore the idea of bringing more flexibility into the traditional convolutions. Spatial Transform Networks [16] predicts a global parametric transformation to warp the feature map, allowing the network to adaptively transform feature maps conditioned on the input. Dynamic filter [17] is proposed to actively predict the parameters of the convolution filters. It applies dynamically generated filters to an image in a sample-specific way. Deformable Convolutional Networks [8] dynamically learn the sampling locations by predicting the offsets for each image location. Pixel-adaptive convolution [29] multiplies the weights of the filters and a spatially varying kernel to make the standard convolution content-adaptive. Yang *et al*. [37] apply conditional *batch normalization* to video object segmentation and AdaptIS [28] predicts the affine parameters, which scale and shift the features conditioned on each instance. They both belong to the more general scale-and-shift operation, which can roughly be seen as an attention mechanism on intermediate feature maps. We bring the dynamic scheme into instance segmentation and enable learning instance segmenters by locations. Note that the concurrent work in [30] also applies dynamic convolutions for instance segmentation by extending the framework of BlendMask [4]. The dynamic scheme part is somewhat similar, but the methodology is different. CondInst [30] relies on the relative position to distinguish instances as in AdaptIS, while SOLOv2 uses absolute positions as in SOLO. It means that it needs to encode the position information $N$ times for $N$ instances, while SOLOv2 performs it all at once using the global coordinates, regardless how many instances there are.

**Non-maximum suppression.** NMS is widely adopted in many computer vision tasks and becomes an essential component of object detection and instance segmentation systems. Some recent works [1, 26, 13, 3, 2] are proposed to improve the traditional NMS. They can be divided into two groups, either for improving the accuracy or speeding up. Instead of applying the hard removal to duplicate predictions according to a threshold, Soft-NMS [1] decreases the confidence scores of neighbors according to their overlap with higher scored predictions. Adaptive NMS [26] applies dynamic suppression threshold to each instance, which is tailored for pedestrian detection in a crowd. In [13], the authors use KL-Divergence and reflected it in the refinement of coordinates in the NMS process. To accelerate the inference, Fast NMS [2] enables deciding the predictions to be kept or discarded in parallel. Note that it speeds up at the cost of performance deterioration. Different from the previous methods, our Matrix NMS addresses the issues of hard removal and sequential operations at the same time. As a result, *the proposed Matrix NMS is able to process 500 masks in less than 1 ms* in simple python implementation, which is negligible compared with the time of network evaluation, and yields 0.4% AP better than Fast NMS.

## 2 Proposed Method: SOLOv2

An instance segmentation system should separate different instances at pixel level. To distinguish instances, we follow the basic concept of 'segmenting objects by locations' [32]. The input image is conceptually divided into $S \times S$ grids. If the center of an object falls into a grid cell, then the grid cell corresponds to a binary mask for that object. As such, the system outputs $S^2$ masks in total, denoted as $M \in \mathbb{R}^{H \times W \times S^2}$. The $k^{th}$ channel is responsible for segmenting instance at position $(i, j)$, where $k = i \cdot S + j$ (see Figure 2(a)).

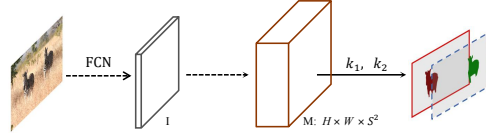

(a) SOLO

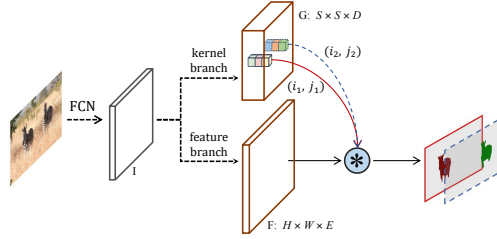

(b) SOLOv2

**Figure 2** – **SOLOv2** compared to SOLO. $I$ is the input feature after FCN-backbone representation extraction. Dashed arrows denote convolutions. $k = i \cdot S + j$; and '⊛' denotes the dynamic convolution operation.

Such paradigm could generate the instance segmentation results in an elegant way. However, there are three main bottlenecks that limit its performance: a) inefficient mask representation and learning. It takes a lot of memory and computation to predict the output tensor $M$, which has $S^2$ channels. Besides, as the $S$ is different for different FPN level, the last layer of each level is learned separately and not shared, which results in an inefficient training. b) inaccurate mask predictions. Finer predictions require high-resolution masks to deal with the details at object boundaries. But large resolutions will considerably increase the computational cost. c) slow mask NMS. Compared with box NMS, mask NMS takes more time and leads to a larger overhead.

In this section, we show that these challenges can be effectively solved by our proposed dynamic mask representation and Matrix NMS, and we introduce them in the sequel.

### 2.1 Dynamic Instance Segmentation

We first revisit the mask generation in SOLO [32]. To generate the instance mask of $S^2$ channels corresponding to $S \times S$ grids, the last layer takes one level of pyramid features $F \in \mathbb{R}^{H \times W \times E}$ as input and at last applies a convolution layer with $S^2$ output channels. The operation can be written as:

$$M_{i,j} = G_{i,j} \circledast F, \tag{1}$$

where $G_{i,j} \in \mathbb{R}^{1 \times 1 \times E}$ is the convolution kernel, and $M_{i,j} \in \mathbb{R}^{H \times W}$ is the final mask containing only one instance whose center is at location $(i, j)$.

In other words, we need two input $F$ and $G$ to generate the final mask $M$. Previous work explicitly output the whole $M$ for training and inference. Note that tensor $M$ is very large, and to directly predict $M$ is memory and computational inefficient. In most cases the objects are located sparsely in the image. $M$ is redundant as only a small part of $S^2$ kernels actually functions during a single inference.

From another perspective, if we separately learn $F$ and $G$, the final $M$ could be directly generated using the both components. In this way, we can simply pick the valid ones from predicted $S^2$ kernels and perform the convolution dynamically. The number of model parameters also decreases. What is more, as the predicted kernel is generated dynamically conditioned on the input, it benefits from the flexibility and adaptive nature. Additionally, each of $S^2$ kernels is conditioned on the location.

It is in accordance with the core idea of segmenting objects by locations and goes a step further by predicting the segmenters by locations.

### 2.1.1 Mask Kernel $G$

Given the backbone and FPN, we predict the mask kernel $G$ at each pyramid level. We first resize the input feature $F_I \in \mathbb{R}^{H_I \times W_I \times C}$ into shape of $S \times S \times C$. Then 4×convs and a final $3 \times 3 \times D$ conv are employed to generate the kernel $G$. We add the spatial functionality to $F_I$ by giving the first convolution access to the normalized coordinates following CoordConv [23], *i.e.*, concatenating two additional input channels which contains pixel coordinates normalized to $[-1, 1]$. Weights for the head are shared across different feature map levels.

For each grid, the kernel branch predicts the $D$-dimensional output to indicate predicted convolution kernel weights, where $D$ is the number of parameters. For generating the weights of a 1×1 convolution with $E$ input channels, $D$ equals $E$. As for 3×3 convolution, $D$ equals $9E$. These generated weights are conditioned on the locations, *i.e.*, the grid cells. If we divide the input image into $S \times S$ grids, the output space will be $S \times S \times D$, There is no activation function on the output.

### 2.1.2 Mask Feature $F$

Since the mask feature and mask kernel are decoupled and separately predicted, there are two ways to construct the mask feature. We can put it into the head, along with the kernel branch. It means that we predict the mask features for each FPN level. Or, to predict a unified mask feature representation for all FPN levels. We have compared the two implementations in Section 3.1.2 by experiments. Finally, we employ the latter one for its effectiveness and efficiency.

For learning a unified and high-resolution mask feature representation, we apply feature pyramid fusion inspired by the semantic segmentation in [18]. After repeated stages of $3 \times 3$ conv, group norm [34], ReLU and 2× bilinear upsampling, the FPN features P2 to P5 are merged into a single output at 1/4 scale. The last layer after the element-wise summation consists of $1 \times 1$ convolution, group norm and ReLU. More details can be referred to supplementary material. It should be noted that we feed normalized pixel coordinates to the deepest FPN level (at 1/32 scale), before the convolutions and bilinear upsamplings. The provided accurate position information is important for enabling position sensitivity and predicting instance-aware features.

### 2.1.3 Forming Instance Mask

For each grid cell at $(i, j)$, we first obtain the mask kernel $G_{i,j,:} \in \mathbb{R}^D$. Then $G_{i,j,:}$ is convolved with $F$ to get the instance mask. In total, there will be at most $S^2$ masks for each prediction level. Finally, we use the proposed Matrix NMS to get the final instance segmentation results.

### 2.1.4 Learning and Inference

The training loss function is defined as follows:

$$L = L_{cate} + \lambda L_{mask}, \tag{2}$$

where $L_{cate}$ is the conventional Focal Loss [21] for semantic category classification, $L_{mask}$ is the Dice Loss for mask prediction. For more details, we refer readers to [32].

During the inference, we forward input image through the backbone network and FPN, and obtain the category score $\mathbf{p}_{i,j}$ at grid $(i, j)$. We first use a confidence threshold of $0.1$ to filter out predictions with low confidence. The corresponding predicted mask kernels are then used to perform convolution on the mask feature. After the `sigmoid` operation, we use a threshold of $0.5$ to convert predicted soft masks to binary masks. The last step is the Matrix NMS.

## 2.2 Matrix NMS

**Motivation.** Our Matrix NMS is motivated by Soft-NMS [1]. Soft-NMS decays the other detection scores as a monotonic decreasing function $f(\texttt{iou})$ of their overlaps. By decaying the scores according to IoUs recursively, higher IoU detections will be eliminated with a minimum score threshold. However, such process is sequential like traditional Greedy NMS and could not be implemented in parallel.

**Table 1** – **Instance segmentation** mask AP (%) on COCO `test-dev`. All entries are *single-model* results. Mask R-CNN* is our improved version with scale augmentation and longer training time (6×). 'DCN' means deformable convolutions used.

|  | backbone | AP | AP$_{50}$ | AP$_{75}$ | AP$_S$ | AP$_M$ | AP$_L$ |
|---|---|---|---|---|---|---|---|
| *box-based:* | | | | | | | |
| Mask R-CNN [12] | Res-101-FPN | 35.7 | 58.0 | 37.8 | 15.5 | 38.1 | 52.4 |
| Mask R-CNN* | Res-101-FPN | 37.8 | 59.8 | 40.7 | **20.5** | 40.4 | 49.3 |
| MaskLab+ [5] | Res-101-C4 | 37.3 | 59.8 | 39.6 | 16.9 | 39.9 | 53.5 |
| TensorMask [6] | Res-101-FPN | 37.1 | 59.3 | 39.4 | 17.4 | 39.1 | 51.6 |
| YOLACT [2] | Res-101-FPN | 31.2 | 50.6 | 32.8 | 12.1 | 33.3 | 47.1 |
| MEInst [38] | Res-101-FPN | 33.9 | 56.2 | 35.4 | 19.8 | 36.1 | 42.3 |
| CenterMask [33] | Hourglass-104 | 34.5 | 56.1 | 36.3 | 16.3 | 37.4 | 48.4 |
| BlendMask [4] | Res-101-FPN | 38.4 | 60.7 | 41.3 | 18.2 | 41.5 | 53.3 |
| *box-free:* | | | | | | | |
| PolarMask [35] | Res-101-FPN | 32.1 | 53.7 | 33.1 | 14.7 | 33.8 | 45.3 |
| SOLO [32] | Res-101-FPN | 37.8 | 59.5 | 40.4 | 16.4 | 40.6 | 54.2 |
| **SOLOv2** | Res-50-FPN | 38.8 | 59.9 | 41.7 | 16.5 | 41.7 | 56.2 |
| **SOLOv2** | Res-101-FPN | 39.7 | 60.7 | 42.9 | 17.3 | 42.9 | 57.4 |
| **SOLOv2** | Res-DCN-101-FPN | **41.7** | **63.2** | **45.1** | 18.0 | **45.0** | **61.6** |

Matrix NMS views this process from another perspective by considering how a predicted mask $m_j$ being suppressed. For $m_j$, its decay factor is affected by: (a) The penalty of each prediction $m_i$ on $m_j$ ($s_i > s_j$), where $s_i$ and $s_j$ are the confidence scores; and (b) the probability of $m_i$ being suppressed. For (a), the penalty of each prediction $m_i$ on $m_j$ could be easily computed by $f(\texttt{iou}_{i,j})$. For (b), the probability of $m_i$ being suppressed is not so elegant to be computed. However, the probability usually has positive correlation with the IoUs. So here we directly approximate the probability by the most overlapped prediction on $m_i$ as

$$f(\texttt{iou}_{\cdot,i}) = \min_{\forall s_k > s_i} f(\texttt{iou}_{k,i}). \tag{3}$$

To this end, the final decay factor becomes

$$decay_j = \min_{\forall s_i > s_j} \frac{f(\texttt{iou}_{i,j})}{f(\texttt{iou}_{\cdot,i})}, \tag{4}$$

and the updated score is computed by $s_j = s_j \cdot decay_j$. We consider two most simple decremented functions, denoted as `linear` $f(\texttt{iou}_{i,j}) = 1 - \texttt{iou}_{i,j}$, and `Gaussian` $f(\texttt{iou}_{i,j}) = \exp\left(-\frac{\texttt{iou}_{i,j}^2}{\sigma}\right)$.

**Implementation.** All the operations in Matrix NMS could be implemented in one shot without recurrence. We first compute a $N \times N$ pairwise IoU matrix for the top $N$ predictions sorted descending by score. For binary masks, the IoU matrix could be efficiently implemented by matrix operations. Then we get the most overlapping IoUs by column-wise max on the IoU matrix. Next, the decay factors of all higher scoring predictions are computed, and the decay factor for each prediction is selected as the most effect one by column-wise min (Eqn. (4)). Finally, the scores are updated by the decay factors. For usage, we just need thresholding and selecting top-$k$ scoring masks as the final predictions.

The pseudo-code of Matrix NMS is provided in supplementary material. In our code base, Matrix NMS is 9×faster than traditional NMS and being more accurate (Table 3(c)). We show that Matrix NMS serves as a superior alternative of traditional NMS in both accuracy and speed, and can be easily integrated into the state-of-the-art detection/segmentation systems.

## 3 Experiments

To evaluate the proposed method SOLOv2, we conduct experiments on three basic tasks, instance segmentation, object detection, and panoptic segmentation on MS COCO [22]. We also present experimental results on the recently proposed LVIS dataset [11], which has more than 1K categories and thus is considerably more challenging.

**Table 2** – Instance segmentation results on the LVISv0.5 validation dataset. ∗ means re-implementation.

| | backbone | $AP_r$ | $AP_c$ | $AP_f$ | $AP_S$ | $AP_M$ | $AP_L$ | AP |
|---|---|---|---|---|---|---|---|---|
| Mask-RCNN [11] | Res-50-FPN | 14.5 | 24.3 | 28.4 | - | - | - | 24.4 |
| Mask-RCNN∗-3× | Res-50-FPN | 12.1 | 25.8 | 28.1 | **18.7** | 31.2 | 38.2 | 24.6 |
| **SOLOv2** | Res-50-FPN | 13.4 | 26.6 | 28.9 | 15.9 | 34.6 | 44.9 | 25.5 |
| **SOLOv2** | Res-101-FPN | **16.3** | **27.6** | **30.1** | 16.8 | **35.8** | **47.0** | **26.8** |

**Table 3** – Ablation experiments for SOLOv2. All models are trained on MS COCO `train2017`, test on `val2017` unless noted.

(a) **Kernel shape.** The performance is stable when the shape goes beyond $1 \times 1 \times 256$.

| Kernel shape | AP | $AP_{50}$ | $AP_{75}$ |
|---|---|---|---|
| $3 \times 3 \times 64$ | 37.4 | 58.0 | 39.9 |
| $1 \times 1 \times 64$ | 37.4 | 58.1 | 40.1 |
| $1 \times 1 \times 128$ | 37.4 | 58.1 | 40.2 |
| $1 \times 1 \times 256$ | **37.8** | **58.5** | **40.4** |
| $1 \times 1 \times 512$ | 37.7 | 58.3 | **40.4** |

(b) **Explicit coordinates.** Precise coordinates input can considerably improve the results.

| Kernel | Feature | AP | $AP_{50}$ | $AP_{75}$ |
|---|---|---|---|---|
| | | 36.3 | 57.4 | 38.6 |
| ✓ | | 36.3 | 57.3 | 38.5 |
| | ✓ | 37.1 | 58.0 | 39.4 |
| ✓ | ✓ | **37.8** | **58.5** | **40.4** |

(c) **Matrix NMS.** Matrix NMS outperforms other methods in both speed and accuracy.

| Method | Iter? | Time(ms) | AP |
|---|---|---|---|
| Hard-NMS | ✓ | 9 | 36.3 |
| Soft-NMS | ✓ | 22 | 36.5 |
| Fast NMS | ✗ | < 1 | 36.2 |
| Matrix NMS | ✗ | < 1 | **36.6** |

(d) **Mask feature representation.** We compare the separate mask feature representation in parallel heads and the unified representation.

| Mask Feature | AP | $AP_{50}$ | $AP_{75}$ |
|---|---|---|---|
| Separate | 37.3 | 58.2 | 40.0 |
| Unified | **37.8** | **58.5** | **40.4** |

(e) **Training schedule.** $1\times$ means 12 epochs using single-scale training. $3\times$ means 36 epochs with multi-scale training.

| Schedule | AP | $AP_{50}$ | $AP_{75}$ |
|---|---|---|---|
| $1\times$ | 34.8 | 54.8 | 36.8 |
| $3\times$ | **37.8** | **58.5** | **40.4** |

(f) **Real-time SOLOv2.** The speed is reported on a single V100 GPU by averaging 5 runs (on COCO `test-dev`).

| Model | AP | $AP_{50}$ | $AP_{75}$ | fps |
|---|---|---|---|---|
| SOLOv2-448 | 34.0 | 54.0 | 36.1 | 46.5 |
| SOLOv2-512 | 37.1 | 57.7 | 39.7 | 31.3 |

## 3.1 Instance Segmentation

For instance segmentation, we report lesion and sensitivity studies by evaluating on the COCO 5K `val2017` split. We also report COCO mask AP on the `test-dev` split, which is evaluated on the evaluation server. SOLOv2 is trained with stochastic gradient descent (SGD). We use synchronized SGD over 8 GPUs with a total of 16 images per mini-batch. Unless otherwise specified, all models are trained for 36 epochs (*i.e.*, $3\times$) with an initial learning rate of 0.01, which is then divided by 10 at 27th and again at 33th epoch. We use scale jitter where the shorter image side is randomly sampled from 640 to 800 pixels.

### 3.1.1 Main Results

We compare SOLOv2 to the state-of-the-art methods in instance segmentation on MS COCO `test-dev` in Table 1. SOLOv2 with ResNet-101 achieves a mask AP of 39.7%, which is much better than other state-of-the-art instance segmentation methods. Our method shows its superiority especially on large objects (*e.g.*, +5.0 $AP_L$ than Mask R-CNN).

We also provide the speed-accuracy trade-off on COCO to compare with some dominant instance segmenters (Figure 1 (a)). We show our models with ResNet-50, ResNet-101, ResNet-DCN-101 and two light-weight versions described in Section 3.1.2. The proposed SOLOv2 outperforms a range of state-of-the-art algorithms, both in accuracy and speed. The running time is tested on our local machine, with a single V100 GPU. We download code and pre-trained models to test inference time for each model on the same machine. Further, as described in Figure 1 (b), SOLOv2 predicts much finer masks than Mask R-CNN which performs on the local region.

Beside the MS COCO dataset, we also demonstrate the effectiveness of SOLOv2 on LVIS dataset. Table 2 reports the performances on the rare (1~10 images), common (11~100), and frequent (> 100) subsets, as well as the overall AP. Both the reported Mask R-CNN and SOLOv2 use data resampling training strategy, following [11]. Our SOLOv2 outperforms the baseline method by about 1% AP. For large-size objects ($AP_L$), our SOLOv2 achieves 6.7% AP improvement, which is consistent with the results on the COCO dataset.

### 3.1.2 Ablation Experiments

We investigate and compare the following five aspects in our methods.

**Kernel shape.** We consider the kernel shape from two aspects: number of input channels and kernel size. The comparisons are shown in Table 3(a). $1 \times 1$ conv shows equivalent performance to $3 \times 3$ conv. Changing the number of input channels from 128 to 256 attains 0.4% AP gains. When it grows beyond 256, the performance becomes stable. In this work, we set the number of input channels to be 256 in all other experiments.

**Effectiveness of coordinates.** Since our method segments objects by locations, or specifically, learns the object segmenters by locations, the position information is very important. For example, if the mask kernel branch is unaware of the positions, the objects with the same appearance may have the same predicted kernel, leading to the same output mask. On the other hand, if the mask feature branch is unaware of the position information, it would not know how to assign the pixels to different feature channels in the order that matches the mask kernel. As shown in Table 3(b), the model achieves 36.3% AP without explicit coordinates input. The results are reasonably good because that CNNs can implicitly learn the absolute position information from the commonly used zero-padding operation, as revealed in [15]. The pyramid zero-paddings in our mask feature branch should have contributed considerably. However, the implicitly learned position information is coarse and inaccurate. When making the convolution access to its own input coordinates through concatenating extra coordinate channels, our method enjoys 1.5% absolute AP gains.

**Unified mask feature representation.** For mask feature learning, we have two options: to learn the feature in the head separately for each FPN level or to construct a unified representation. For the former one, we implement as SOLO and use seven $3 \times 3$ convolutions to predict the mask features. For the latter one, we fuse the FPN's features in a simple way and obtain the unified mask representations. The detailed implementation is in supplementary material. We compare these two modes in Table 3(d). As shown, the unified representation achieves better results, especially for the medium and large objects. This is easy to understand: In separate way, the large-size objects are assigned to high-level feature maps of low spatial resolutions, leading to coarse boundary prediction.

**Matrix NMS.** Our Matrix NMS can be implemented totally in parallel. Table 3(c) presents the speed and accuracy comparison of Hard-NMS, Soft-NMS, Fast NMS and our Matrix NMS. Since all methods need to compute the IoU matrix, we pre-compute the IoU matrix in advance for fair comparison. The speed reported here is that of the NMS process alone, excluding computing IoU matrices. Hard-NMS and Soft-NMS are widely used in current object detection and segmentation models. Unfortunately, both methods are recursive and spend much time budget (*e.g.*, 22 ms). Our Matrix NMS only needs < 1 ms and is almost cost free! Here we also show the performance of Fast NMS, which utilizes matrix operations but with performance penalty. To conclude, our Matrix NMS shows its advantages on both speed and accuracy.

**Real-time setting.** We design two light-weight models for different purposes. 1) `Speed priority`, the number of convolution layers in the prediction head is reduced to two and the input shorter side is 448. 2) `Accuracy priority`, the number of convolution layers in the prediction head is reduced to three and the input shorter side is 512. Moreover, deformable convolution [8] is used in the backbone and the last layer of prediction head. We train both models with the $3\times$ schedule, with shorter side randomly sampled from [352, 512]. Results are shown in Table 3(f). SOLOv2 can not only push state-of-the-art, but has also been ready for real-time applications.

### 3.2 Extensions: Object Detection and Panoptic Segmentation

Although our instance segmentation solution removes the dependence of bounding box prediction, we are able to produce the 4-D object bounding box from each instance mask. The best model of ours achieve 44.9% AP on COCO `test-dev`. SOLOv2 beats most recent methods in both accuracy and speed, as shown in Figure 3. Here we emphasize that our results are directly generated from the off-the-shelf instance mask, without any box based supervised training or engineering.

Besides, we also demonstrate the effectiveness of SOLOv2 on the problem of panoptic segmentation. The proposed SOLOv2 can be easily extended to panoptic segmentation by adding the semantic segmentation branch, analogue to the mask feature branch. We use annotations of COCO 2018 panoptic segmentaiton task. All models are trained on `train2017` subset and tested on `val2017`.

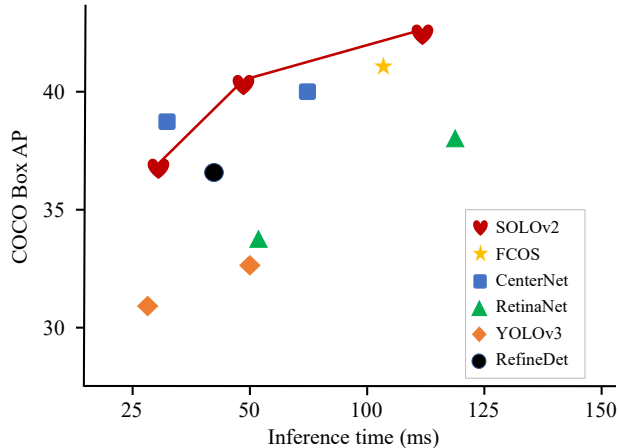

**Figure 3** – SOLOv2 for **object detection**. Speed-accuracy trade-off of bounding-box detection on the COCO `test-dev`.

**Table 4** – SOLOv2 for **panoptic segmentation** – results on COCO val2017. $*$ means re-implementation.

|  | PQ | $PQ^{Th}$ | $PQ^{St}$ |
|---|---|---|---|
| *box-based:* |  |  |  |
| AUNet [19] | 39.6 | 49.1 | 25.2 |
| UPSNet [36] | **42.5** | 48.5 | **33.4** |
| Panoptic-FPN [18] | 39.0 | 45.9 | 28.7 |
| Panoptic-FPN$^*$-1$\times$ | 38.7 | 45.9 | 27.8 |
| Panoptic-FPN$^*$-3$\times$ | 40.8 | 48.3 | 29.4 |
| *box-free:* |  |  |  |
| AdaptIS [28] | 35.9 | 40.3 | 29.3 |
| SSAP [10] | 36.5 | — | — |
| Pano-DeepLab [7] | 39.7 | 43.9 | **33.2** |
| **SOLOv2** | **42.1** | **49.6** | 30.7 |

We use the same strategy as in Panoptic-FPN to combine instance and semantic results. As shown in Table 4, our method achieves state-of-the-art results and outperforms other recent box-free methods by a large margin. All methods listed use the same backbone (ResNet50-FPN) except SSAP (ResNet101) and Pano-DeepLab (Xception-71). Note that UPSNet has used deformable convolution [8] for better performance.

# 4 Conclusion

In this paper, we proposed SOLOv2, a dynamic and fast instance segmentation solution with strong performance. The method includes three key techniques. *a)* We proposed to learn dynamic convolutional kernels for the mask prediction, conditioned on the location, which leads to a much more compact yet more powerful head design, and achieving better results; *b)* We re-designed the object mask generation in a simple and unified way, which yields more accurate boundaries; *c)* Moreover, unlike box NMS as in object detection, for direct instance segmentation a bottleneck in inference efficiency is the NMS of masks. We developed a simple and much faster NMS strategy, termed Matrix NMS, for NMS processing of masks, without sacrificing mask AP.

Our experiments on the MS COCO and LVIS datasets demonstrate the superior performance in terms of both accuracy and speed of the proposed SOLOv2. Being versatile for instance-level recognition tasks, we show that without any modification to the framework, SOLOv2 performs competitively for panoptic segmentation. Thanks to its simplicity (being proposal free, anchor free, FCN-like), strong performance in both accuracy and speed, and potentially being capable of solving many instance-level tasks, SOLOv2 can be a strong baseline approach to instance recognition and beyond.

## Acknowledgement

Tao Kong and Chunhua Shen are the corresponding authors. Chunhua Shen and his employer received no financial support for the research, authorship, and/or publication of this article. We would like to thank Mingxuan Jing, Yuchen Mo, and Ruihang Chu for their valuable comments and suggestions.

## Broader Impact

One of the primary goals of computer vision is understanding of visual scenes. Scene understanding involves numerous tasks (*e.g.*, recognition, detection, segmentation, *etc*.). Among them, instance segmentation is probably one of the most challenging tasks, which requires to detect object instances at the pixel level.

Albeit being challenging, instance segmentation is beneficial to a wide range of applications, including autonomous driving, augmented reality, medical image analysis, and image/video editing. The proposed accurate and fast instance segmentation solution benefits broader applications. Autonomous driving becomes safer. Doctors could find the lesion part in medical images with less effort.

Moreover, we believe that our method can serve as a strong baseline for researchers and engineers in the field. This new paradigm may encourage future work to deeply analyze and further enhance research along this direction. Practitioners may develop interesting applications built upon our approach.

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
