[Supplementary Material]

# SOLOv2: Dynamic and Fast Instance Segmentation Appendix

**Xinlong Wang**[1]      **Rufeng Zhang**[2]      **Tao Kong**[3]      **Lei Li**[3]      **Chunhua Shen**[1]

[1] The University of Adelaide, Australia      [2] Tongji University, China      [3] ByteDance AI Lab

## A  Matrix NMS

The pseudo-code of Matrix NNS is shown in Figure 1. All the operations in Matrix NMS could be implemented in one shot without recurrence. In our code base, Matrix NMS is $9\times$ times faster than traditional NMS and being more accurate. We show that Matrix NMS serves as a superior alternative of traditional NMS both in accuracy and speed, and can be easily integrated into the state-of-the-art detection/segmentation systems.

```python
def matrix_nms(scores, masks, method='gauss', sigma=0.5):
    # scores: mask scores in descending order (N)
    # masks: binary masks (NxHxW)
    # method: 'linear' or 'gauss'
    # sigma: std in gaussian method

    # reshape for computation: Nx(HW)
    masks = masks.reshape(N, HxW)
    # pre-compute the IoU matrix: NxN
    intersection = mm(masks, masks.T)
    areas = masks.sum(dim=1).expand(N, N)
    union = areas + areas.T - intersection
    ious = (intersection / union).triu(diagonal=1)

    # max IoU for each: NxN
    ious_cmax = ious.max(0)
    ious_cmax = ious_cmax.expand(N, N).T
    # Matrix NMS, Eqn.(4): NxN
    if method == 'gauss': # gaussian
        decay = exp(-(ious^2 - ious_cmax^2) / sigma)
    else: # linear
        decay = (1 - ious) / (1 - ious_cmax)
    # decay factor: N
    decay = decay.min(dim=0)
    return scores * decay
```

Figure 1: Python code of Matrix NMS. `mm`: matrix multiplication; `T`: transpose; `triu`: upper triangular part

## B  Unified Mask Feature Representation

The detailed implementation is illustrated in Figure B. For learning a unified and high-resolution mask feature representation, we apply feature pyramid fusion inspired by the semantic segmentation in [1]. After repeated stages of $3 \times 3$ conv, group norm [2], ReLU and $2\times$ bilinear upsampling, the FPN features P2 to P5 are merged into a single output at 1/4 scale. The last layer after the element-wise summation consists of $1 \times 1$ convolution, group norm and ReLU. It should be noted that we feed normalized pixel coordinates to the deepest FPN level (at 1/32 scale), before the convolutions and bilinear upsamplings. The provided accurate position information is important for enabling position sensitivity and predicting instance-aware features. Compared with the separated alternative, the unified mask feature representation is more effective and time efficient.

Figure 2: **Unified mask feature branch**. Each FPN level (left) is upsampled by convolutions and bilinear upsampling until it reaches 1/4 scale (middle). In the deepest FPN level, we concatenate the $x$, $y$ coordinates and the original features to encode spatial information. After element-wise summation, a $1 \times 1$ convolution is attached to transform to designated output mask feature $F \in \mathbb{R}^{H \times W \times E}$.

## C  Visualization

We visualize what our SOLOv2 has learnt from two aspects: mask feature behavior and the final outputs after being convolved by the dynamically learned convolution kernels.

We visualize the outputs of mask feature branch. We use a model which has 64 output channels (*i.e.*, $E = 64$ for the last feature map prior to mask prediction) for easy visualization. Here we plot each of the 64 channels (recall the channel spatial resolution is $H \times W$) as shown in Figure 3.

There are two main patterns. The first and the foremost, the mask features are position-aware. It shows obvious behavior of scanning the objects in the image horizontally and vertically. The other obvious pattern is that some feature maps are responsible for activating all the foreground objects, *e.g.*, the one in white boxes.

The final outputs are shown in Figure 5. Different objects are in different colors. Our method shows promising results in diverse scenes. It is worth pointing out that the details at the boundaries are segmented well, especially for large objects.

We also provide three videos for better visualization of our instance segmentation results. These videos are generated from frame-by-frame inference, without any temporal processing. Though only trained on MS COCO, our model generalizes well across various scenes.

Figure 3: **Mask feature behavior.** Each plotted subfigure corresponds to one of the 64 channels of the last feature map prior to mask prediction. The mask features appear to be position-sensitive (orange box), while a few mask features are position-agnostic and activated on all instances (white box). Best viewed on screens.

# D   Bounding-box Object Detection

Although our instance segmentation solution removes the dependence of bounding box prediction, we are able to produce the 4D object bounding box from each instance mask. In Table 1, we compare the generated box detection performance with other object detection methods on COCO. All models are trained on the `train2017` subset and tested on `test-dev`.

As shown in Table 1, our detection results outperform most methods, especially for objects of large scales, demonstrating the effectiveness of SOLOv2 in object box detection. Similar to instance segmentation, we also plot the speed/accuracy trade-off curve for different methods in Figure 4. We show our models with ResNet-101 and two light-weight versions described above. The plot reveals that the bounding box performance of SOLOv2 beats most recent object detection methods in both accuracy and speed. Here we emphasis that our results are directly generated from the off-the-shelf instance mask, without any box based supervised training or engineering.

An observation from Figure 4 is as follows. If one does not care much about the cost difference between mask annotation and bounding box annotation, it appears to us that there is no reason to use box detectors for downstream applications, considering the fact that our SOLOv2 beats most modern detectors in both accuracy and speed.

|  | backbone | AP | $AP_{50}$ | $AP_{75}$ | $AP_S$ | $AP_M$ | $AP_L$ |
|---|---|---|---|---|---|---|---|
| YOLOv3 [3] | DarkNet53 | 33.0 | 57.9 | 34.4 | 18.3 | 35.4 | 41.9 |
| SSD513 [4] | ResNet-101 | 31.2 | 50.4 | 33.3 | 10.2 | 34.5 | 49.8 |
| DSSD513 [4] | ResNet-101 | 33.2 | 53.3 | 35.2 | 13.0 | 35.4 | 51.1 |
| RefineDet [5] | ResNet-101 | 36.4 | 57.5 | 39.5 | 16.6 | 39.9 | 51.4 |
| Faster R-CNN [6] | Res-101-FPN | 36.2 | 59.1 | 39.0 | 18.2 | 39.0 | 48.2 |
| RetinaNet [7] | Res-101-FPN | 39.1 | 59.1 | 42.3 | 21.8 | 42.7 | 50.2 |
| FoveaBox [8] | Res-101-FPN | 40.6 | 60.1 | 43.5 | 23.3 | 45.2 | 54.5 |
| RPDet [9] | Res-101-FPN | 41.0 | 62.9 | 44.3 | 23.6 | 44.1 | 51.7 |
| FCOS [10] | Res-101-FPN | 41.5 | 60.7 | 45.0 | **24.4** | 44.8 | 51.6 |
| CenterNet [11] | Hourglass-104 | 42.1 | 61.1 | 45.9 | 24.1 | 45.5 | 52.8 |
| **SOLOv2** | Res-50-FPN | 40.4 | 59.8 | 42.8 | 20.5 | 44.2 | 53.9 |
| **SOLOv2** | Res-101-FPN | 42.6 | 61.2 | 45.6 | 22.3 | 46.7 | 56.3 |
| **SOLOv2** | Res-DCN-101-FPN | **44.9** | **63.8** | **48.2** | 23.1 | **48.9** | **61.2** |

Table 1: **Object detection** box AP (%) on the COCO `test-dev`. Although our bounding boxes are directly generated from the predicted masks, the accuracy outperforms most state-of-the-art methods. Speed-accuracy trade-off of typical methods is shown in Figure 4.

Figure 4: Speed-accuracy trade-off of bounding-box **object detection** on the COCO `test-dev`.

Figure 5: **Visualization of instance segmentation results** using the Res-101-FPN backbone. The model is trained on the COCO `train2017` dataset, achieving a mask AP of 39.7% on the COCO `test-dev`.