[Reviews · NeurIPS 2020]

Review 1

Summary and Contributions: The paper proposes a novel approach for fast and robust instance segmentation. The authors propose to apply dynamically generated kernel weights for mask generation to a novel instance segmentation idea, segmenting by location (proposed in SOLO). The resulting network architecture avoids the inefficiency of mask representation in the original SOLO paper. The dynamic kernel weight also improves the overall model accuracy. The authors also propose an efficient mask NMS post-processing algorithm that further speeds up the inference process. Evaluation of the proposed approach is done on benchmark datasets and the model is able to outperform the state-of-the-art in both accuracy and inference efficiency.

Strengths: Although the mask representation is not new, the solution to better utilize this representation is novel and valuable. The performance improvement on the COCO-test is solid. The ablation analysis is sufficient and beneficial to the community. The idea of decoupling the learning of feature maps and the kernels to decrease the final output feature dimension could potentially inspire other applications.

Weaknesses: I think the experiment section could be improved to make the submission stronger. The author indicated that evaluation is done on COCO and LVIS in the main text while the result table of LVIS is only included in the supplemental material. I think it is worth to bring them all in the main paper. For panoptic segmentation results, I think it is misleading that UPSNet is not included in the main paper but included in the supplemental material. As panoptic segmentation is not the main targeted application, it wouldn’t change the whole story to include UPSNet. One can still leave the note on the usage of DCN, which might be responsible for better performance.

Correctness: Yes, the method is correct and the claims are sounded.

Clarity: The paper is well-formatted and written, easy to follow, and understand.

Relation to Prior Work: The literature reivew is good in general. The comparison to the closest related work, SOLO is also thoroughfully done. Some missing / correction on citations that doesn’t require experiment updates: 1. Pano-deeplab is accepted by CVPR 2020:Bowen Cheng, Maxwell D. Collins, Yukun Zhu, Ting Liu, Thomas S. Huang, Hartwig Adam, Liang-Chieh Chen; Panoptic-DeepLab: A Simple, Strong, and Fast Baseline for Bottom-Up Panoptic Segmentation, Proceedings of the IEEE/CVF Conference on Computer Vision and Pattern Recognition (CVPR), 2020, pp. 12475-12485 2. Dynamic convolution: Su, H., Jampani, V., Sun, D., Gallo, O., Learned-Miller, E. and Kautz, J., 2019. Pixel-adaptive convolutional neural networks. In Proceedings of the IEEE Conference on Computer Vision and Pattern Recognition (pp. 11166-11175). 3. Novel Mask representation: Hou, R., Li, J., Bhargava, A., Raventos, A., Guizilini, V., Fang, C., Lynch, J. and Gaidon, A., 2020. Real-Time Panoptic Segmentation from Dense Detections. In Proceedings of the IEEE/CVF Conference on Computer Vision and Pattern Recognition (pp. 8523-8532). 4. Novel Mask representation: Yang, T.J., Collins, M.D., Zhu, Y., Hwang, J.J., Liu, T., Zhang, X., Sze, V., Papandreou, G. and Chen, L.C., 2019. Deeperlab: Single-shot image parser. arXiv preprint arXiv:1902.05093. (FYI since this is not published in peer review conference yet) 5. NMS: Lile Cai, Bin Zhao, Zhe Wang, Jie Lin, Chuan Sheng Foo, Mohamed Sabry Aly, Vijay Chandrasekhar; MaxpoolNMS:Getting Rid of NMS Bottlenecks in Two-Stage Object Detectors Proceedings of the IEEE/CVF Conference on Computer Vision and Pattern Recognition (CVPR), 2019, pp. 9356-9364 6. Use pixel coordinate/location as input: Liu, R., Lehman, J., Molino, P., Such, F.P., Frank, E., Sergeev, A. and Yosinski, J., 2018. An intriguing failing of convolutional neural networks and the coordconv solution. In Advances in Neural Information Processing Systems (pp. 9605-9616).

Reproducibility: Yes

Additional Feedback: Other minor suggestions: In the abstract, please indicate dataset along with AP. Eq. 3, use math font for ‘iou’ on the left for consistency. After rebuttal ------------------ Thanks for addressing my concerns. However, after reading all the other reviews, I agree that the techinical contributions are not original enough for a clear accept. Most of the contributions replies on apply existent techinques to address specific problems in SOLO. But I still believe the submission is good for publication as the overall methods presents a more efficient instance segmentation and will benefit the community.


Review 2

Summary and Contributions: - This paper improves upon SOLO in two aspects. First, mask prediction is decoupled into a dynamic mask generation and mask feature learning, avoiding the need to produce masks w.r.t. all S^2 positions as in SOLO. - Second, the authors introduce Matrix NMS which significantly speeds up the post-processing step. - Extensive experimental results demonstrate the effectiveness of DFIS in term of both speed and accuracy.

Strengths: - Writing is clear and easy to follow. - Overall idea is simple yet very effective. Extensive experiments including ablation study and extension on other tasks (e.g. object detection and panoptic segmentation) clearly demonstrate the effectiveness of the proposed solution. - I think the proposed idea is practically useful and could benefit future works on this area.

Weaknesses: - Despite the effectiveness of the proposed approach as proven by extensive experiments, the technical novelties are fairly limited. Applying dynamic operations for dynamic mask generation has been previously explored in the segmentation literature, e.g. conditional batch normalization [1], adaptive instance normalization [2], but are not discussed here. What is the benefit of using dynamic convolution as opposed to other conditional operations? - The Fig.4 in SOLO illustrates the location-awareness of each learned kernel, i.e. different instance is activated at different channel (among S^2 channels) depending on the position of the grid in the image. It would be interesting to also provide a similar plot for DFIS (each subfigure corresponds to the mask produced by one of the S^2 kernels) to study whether such position-aware property still holds in DFIS. - Eqn(1) should be M_i,j = G_i,j * F instead. - Typo: threshing in L207 References: [1] Sofiiuk et al. “AdaptIS: Adaptive instance selection network”, in ICCV 2019. [2] Yang et al. “Efficient video object segmentation via network modulation”, in CVPR 2018.

Correctness: Yes

Clarity: Yes

Relation to Prior Work: Yes

Reproducibility: Yes

Additional Feedback: There is a concurrent work [3] (accepted by ECCV 2020) which also applies dynamic convolution in a similar way for instance segmentation. Authors could consider citing and explain the difference between their work and [3]. Reference: [3] Tian et al. “Conditional convolution for instance segmentation”, in ECCV 2020. --------------------------------------------------------------------------------------------------------- I think the rebuttal addressed all my concerns. After going through the other reviewers' comments and authors' response, I decide to keep my rating. Although the overall idea is fairly similar to CondInst [3], the proposed method is practically useful and its remarkable performance makes it a strong baseline for instance segmentation.


Review 3

Summary and Contributions: This paper aims at building a simple, direct and fast framework for instance segmentation. To this end, the authors proposes an effective approach, termed Dynamic and Fast Instance Segmentation (DFIS). Specifically, the object mask generation is decoupled into a mask kernel prediction and mask feature learning. Furthermore, a novel matrix non-maximum suppression (NMS) technique is introducted to significantly reduces inference overhead. Extensive experiments on object detection and panoptic segmentation tasks show the superiority of the proposed method.

Strengths: + The motivation of this paper is solid, and the paper is well written + The idea of using dynamic convolutions is interesting and reasonable. + Extensive experiments are conducted to verify the effectiveness and efficiency of the proposed method. + The experiments are sufficient.

Weaknesses: First of all, the idea of Dynamic Instance Segmentation is very similar to [1,2]. Both this paper and [1] utilize dynamic convolutions to generate mask kernels for further segmentation, while [2] adopt the AdaIN as an alternative. All of them use coordinate maps as a condition. Second, both this paper and [1] use a unified maks feature representation for all FPN levels. Could the author please clarifry the difference ? - The coordinate information seems to be important according to Tab. 2(b). However, the details of two additional coordinate channels in both mask kernel G and mask feature F are not very clear. - Inference of the propsoed methods is very efficient. However, in Tab. 2(c), the author only discussed the efficiency of Matrix NMS. How about the contribution of dynamic mask representation in terms of speed? Typos: - L229, espically->especially - L287, emphasis ->emphasize - L323, could becomes -> could become [1] Conditional Convolutions for Instance Segmentation. ECCV2020 [2] AdaptIS: Adaptive Instance Selection Network. ICCV 2019

Correctness: The claims and empirical methodology seem to be correct.

Clarity: Yes

Relation to Prior Work: See weakness.

Reproducibility: Yes

Additional Feedback: The authors are encouraged to discuss the problems mentioned in weaknesses.


Review 4

Summary and Contributions: The author proposed an improved SOLO instance segmentation method, which address three issues in the original SOLO, namely storage, accuracy and NMS of mask prediction.

Strengths: The idea is reasonable. The proposed method is effective and efficient. Though largely based on SOLO, the modification of mask generation and mask NMS are non-trivial. The experiments on COCO are rather thorough.

Weaknesses: Though I think this work has done a great job on the experiment side on COCO, the major problem of this paper is the writing. 1. No need to predict the bounding box is a feature but not a success. For one thing, bounding box does coarse and unnatural (L24) for shape representation, but it is an intermediate path to balance human labor and money cost for annotation, and enough for many real-world applications. “with our … solutions, instance segmentation should be an advanced alternative to … object bounding box detection” at Line 65 is a little bit arrogant. Altering the algorithm is easy, but preparing the annotation is expensive. Besides, proposal-free/bottom-up instance segmentation is a long-standing research line, the proposed method belongs to this line, thus naturally do not need to predict the bounding box. 2. Based on 1 I mention above, it seems the author does not realize the really close relationship between the proposed method to bottom-up line of instance segmentation, and “specifically compare our method with the recent YOLACT” (at line 79), on what ground? real-time efficiency? Though a bunch of bottom-up method only train and test on the cityscapes dataset, but it is also a challenging dataset for the amount of the small objects. According to Table 1, even the strongest DFIS cannot beat the Mask R-CNN with AP_S. But “Instance Segmentation by Jointly Optimizing Spatial Embeddings and Clustering Bandwidth” in CVPR 2019 can already beat the Mask R-CNN on the small objects in cityscapes dataset. 3. To describe the network structure, figures instead of text can really help understanding 4. s_i, s_j at line 194 come from no-where. And since the formulation of Matrix NMS is based on them, which makes the formulation incomprehensible. It may be a notion from Soft NMS, but it should be clearly pointed out. I do have divisive opinions about this paper, for one thing, though COCO is not a perfect benchmark here in my opinion, it is indeed the widely accepted one, for another, I do think the paper writing should be significantly enhanced. Given the responsibility of a reviewer is to help polish the paper, I think improving the writing in the camera-ready cannot be assured. Thus I lean to take the writing side.

Correctness: Yes

Clarity: The structure of paper is vanished in too many details.

Relation to Prior Work: Yes

Reproducibility: Yes

Additional Feedback:

[Author Response · NeurIPS 2020]

We thank all the reviewers for their constructive comments, which are addressed as follows.

To Reviewer 1:

**Q1: Move experiments from supp. to the main paper.** Thanks for this suggestion, and we will move the LVIS
instance segmentation results and COCO panoptic segmentation results from supplementary to the main paper.

**Q2: Missing citations.** Thanks for your suggestion. The suggested citations will be added.

To Reviewer 2:

**Q1: Comparisons with other dynamic operations [1, 2].** In video object seg-
mentation, Yang *et al.*[2] apply conditional *batch normalization* to manipulate the
intermediate feature maps, making the feature map focus on a specific object instance.
Similarly, AdaptIS [1] predicts the affine parameters, which scale and shift the fea-
tures conditioned on each instance. It requires a large mask head to achieve good
results. Both the operations used in [1, 2] belong to the more general scale-and-shift
operation, which can roughly be seen as an attention mechanism on intermediate
feature maps. In contrast, the dynamic convolution naturally suits our purpose of
dynamic mask representation, as it directly decouples the object mask generation
into mask kernel prediction and mask feature learning. Quantitatively, we compared
with [1] in Table 3 of supplementary, where DFIS shows 6.2 $PQ$ and 9.3 $PQ^{things}$
better than [1]. We should add the above discussions to the related work.

Figure A: Position-aware prop-
erty. The duplicated masks will
be suppressed by the proposed
Matrix NMS.

**Q2: Visualization of position-aware property in DFIS.** We show an example in
Figure A. More visualization will be provided.

**Q3: Comparison with concurrent work CondInst [3].** The dynamic scheme part
is somewhat similar, as they both are inspired by Dynamic Filter Networks [Brabandere et al. NIPS'16]. But the
methodology is different. (a) [3] relies on the relative position to distinguish instances as in AdaptIS, while DFIS uses
absolute positions as in SOLO. It means that [3] needs to encode the position information $N$ times for $N$ instances,
while DFIS performs it all at once using the global coordinates, regardless how many instances there are. (b) [3]
works in a two-stage manner that first performs box detection and then segments the detected objects. In contrast, our
proposed DFIS is much simpler, which takes an image as input, directly outputs instance masks and corresponding
class probabilities. For example, [3] has at least 4 loss terms while DFIS has 2 loss terms. (c) In addition, our Matrix
NMS serves as an important contribution.

To Reviewer 3:

**Q1: Comparisons with AdaptIS and CondInst.** We address the concerns of the comparisons with AdaptIS [1] and
the concurrent work CondInst [3] in R2.Q1 and R2.Q3.

**Q2: The details of coordinate channels.** A tensor of the same spatial size as input is created, which contains pixel
coordinates normalized to $[-1, 1]$. We then concatenate it to the input features and feed to the following layers.
Specifically, given the $H \times W \times D$ shaped feature tensor, the size of the new tensor is $H \times W \times (D + 2)$, where the
last two channels are $x$-$y$ pixel coordinates.

**Q3: The contribution of dynamic mask representation in terms of speed.** Using the same NMS strategy, the
Res-101 DFIS runs at 15.1 FPS on a V100 GPU vs. SOLO's 11.6 FPS. The speed-up mostly comes from fewer convs
(7 to 4), as the dynamic mask representation enables us to achieve superior results with lighter prediction head.

To Reviewer 4:

**Q1: About the claim of 'no need to predict the bounding box'.** We agree that the bounding box serves as an
effective and efficient representation for object localization. We are not saying that BBox object detection is not
important. The point is that our instance segmentation method doesn't reply on BBox prediction. We will remove
"...instance segmentation should be an advance alternative..." in L65 to eliminate the confusion.

**Q2: The relationship between DFIS and bottom-up instance segmentation.** We agree that the bottom-up method
is a long-standing research line and has shown competitive results on some datasets, *e.g.*, Cityscapes. But the key
difference here is that the bottom-up method needs post-processing to group pixels into individual object masks, while
DFIS directly predicts explicit object masks, without the restriction of pixel grouping.

**Q3: Others.** L65 will be removed. About the L79, the comparison is on methodology. The quantitative comparison
including real-time efficiency is shown in Figure 1(a) and Table 1. About L194, thanks for pointing it out. $s_i$ and $s_j$ are
the confidence scores. All above explanations will be added to our paper to make it more clear.

[Meta-Review · NeurIPS 2020]

The reviewers found that this paper provides excellent results for image segmentation, building off of previous work. There was some disagreement among the reviewers on a few areas. The first of these is novelty -- always a tricky subject -- where some reviewers noted that the work appears to be incremental methodologically. However, given that the results provide a novel contribution to the field, I think that this is not a particularly strong drawback. There were questions raised about the clarity of the paper. I believe that these have been reasonably addressed by the author response -- and I have also read the paper myself to judge clarity, and feel that it is acceptable. That said, I do expect the authors to fully incorporate the feedback from the reviewers to further improve the paper, as promised in their author response.